# Clinical spectrum of endemic leptospirosis in relation to cytokine response

Niroshana J. Dahanayaka[1,2☯], Suneth B. Agampodi[3,4☯]*, Indika Seneviratna[5], Janith Warnasekara[4], Rukman Rajapakse[1], Kosala Ranathunga[1], Michael Matthias[3], Joseph M. Vinetz[3]

1 Faculty of Medicine and Allied Sciences, Department of Medicine, Rajarata University of Sri Lanka, Saliyapura, Sri Lanka, 2 Faculty of Medicine, Department of Medicine, University of Ruhuna, Matara, Sri Lanka, 3 Department of Internal Medicine, Section of Infectious Diseases, School of Medicine, Yale University, New Haven, Connecticut, United States of America, 4 Faculty of Medicine and Allied Sciences, Department of Community Medicine, Rajarata University of Sri Lanka, Saliyapura, Sri Lanka, 5 Faculty of Medicine and Allied Sciences, Department of Biochemistry, Rajarata University of Sri Lanka, Saliyapura, Sri Lanka

☯ These authors contributed equally to this work.
* suneth.agampodi@yale.edu, sunethagampodi@yahoo.com

## Abstract

### Objectives

To describe the clinical spectrum and the cytokine response of leptospirosis patients in an endemic setting of Sri Lanka.

### Methods

Patients presenting to the university teaching hospital, Anuradhapura, Sri Lanka with a leptospirosis-compatible illness were recruited over a period of 12 months starting from June 2012. Daily clinical and biochemical parameters of the patients were prospectively assessed with a follow-up of 14 days after discharge. A magnetic bead–based multiplex cytokine kit was used to detect 17 cytokines.

### Results

Of the 142 clinically suspected leptospirosis patients recruited, 47 were confirmed and, 29 cases were labeled as "probable." Thrombocytopenia and leukocytosis were observed at least once during the hospital stay among 76(54%) and 39(28%) patients, respectively. Acute kidney injury was observed in 31 patients (22%) and it was significantly higher among confirmed and probable cases. Hu TNF-α and IL-1β were detected only in patients without complications. Hu MIP-1b levels were significantly higher among patients with complications. During the convalescence period, all tested serum cytokine levels were lower compared to the acute sample, except for IL-8. The cytokine response during the acute phase clustered in four different groups. High serum creatinine was associated GM-CSF, high IL-5 and IL-6 level were correlates with lung involvement and saturation drop. The patients with high billirubin (direct)>7 mmol/l had high IL-13 levels.

**Data Availability Statement:** All relevant data are within the manuscript and its Supporting Information files.

**Funding:** This work is partially funded through Rajarata University of Sri Lanka Research Grant # RJT / RP&HDC / 2014 /FMAS /R / 03. SBA, MM and JV are supported by NIAID of the National Institutes of Health under award number U19AI115658.The funders had no role in study design, data collection and analysis, decision to publish, or preparation of the manuscript.

**Competing interests:** No authors have competing interests.

## Conclusions

Results of this study confirms that the knowledge on cytokine response in leptospirosis could be more complex than other similar tropical disease, and biosignatures that provide diagnostic and prognostic information for human leptospirosis remain to be discovered.

## Introduction

Globally, 2.9 million disability-adjusted life years per annum are estimated to be lost owing to leptospirosis [1]. This burden results in more than 1 million estimated cases and 58,900 deaths [2]. Global studies in recent years have promoted awareness of leptospirosis and at the same time increased the demand for high-quality data on which to design better interventions to prevent and better treat this disease. Despite the accumulating evidence on various aspects of leptospirosis, its control, prevention and clinical management remain as major challenges for clinicians and public health professionals in tropical countries where the disease burden is highest.

Clinical studies on leptospirosis have revealed a wide variation of morbidity and mortality. A recent systematic review showed that, among untreated patients, median mortality ranges from 2% to 37% among diagnosed cases [3]. With appropriate treatment—requiring prompt diagnosis mortality was much lower and outcomes better. Based on the study setting, country, selection criteria, and infecting strains, the clinical syndrome presented by most patients varies between typical Weil's disease and self-resolving undifferentiated fever. Understanding the typical disease pattern is often difficult in terms of disease burden estimates, owing to sampling biases [3].

Leptospirosis is hyperendemic in Sri Lanka. Since 2008 [4], incidence based on routine notification exceeds 4,000 cases per year. Moreover, since 2003, prospective studies from Kandy [5], Colombo [6,7], Gampaha [6], Galle [8], Peradeniya [9,10], Kegalle [11], and Anuradhapura [12] have reported different rates of severe disease, varying case fatality rates, and a wide range of clinical manifestations. These differences have been attributed, at least in part to micro-geographical differences [12]; molecular characterization has demonstrated that infecting *Leptospira* different in different ecological and socio-demographic contexts [7,9,11,12].

The different disease outcomes in leptospirosis could be attributed to various factors such as; presence of virulence factors in certain serovar [13] or difference in host immune response [14] such as patterns of cytokine production in early phase of the infection. IL-6, IL-8, IL-10 TNF-$\alpha$ and long pentraxin-3 (PTX3) are shown to be closely related to disease severity and mortality [14–17]. However, Mikulski *et al* showed that TNF-$\alpha$, IL-β, IL-1γa, PTX3, IL-6 and IL-10 levels between mild and severe cases are not significantly differ [18]. Contradictory results are available for IL-10/TNF-$\alpha$ ratio [17,18]. Bandara et al shows an involvement of Th17 cells in the immunopathogenesis of leptospirosis with an increase in IL-17A, IL-21 and IL-23 levels [19]. However, the cytokine response reported in different studies were shown to be not conclusive and more data are required to have conclusive evidence [20].

To better understand the various aspects of clinical leptospirosis, robust data based on prospective studies with minimal sampling bias need to be carried out. The present study aimed to describe the full clinical spectrum and laboratory findings (including cytokine response) of a prospectively studied, systematically sampled cohort of patients from a single clinical research unit over a 1-year period in Anuradhapura, Sri Lanka.

## Materials and methods

### Study setting

This prospective, longitudinal cohort study included subjects based on presenting with a leptospirosis-compatible illness, beginning at the time of each patient's hospital admission and continuing through outpatient follow-up after discharge up to 14 days. The study was carried out in the Faculty of Medicine and Allied Sciences teaching unit of the teaching hospital, Anuradhapura (THA) affiliated with Rajarata University of Sri Lanka. Anuradhapura district typically reports 100–150 cases of leptospirosis annually, with the majority of cases reported from THA. THA, the only sentinel site in Anuradhapura district for leptospirosis surveillance, has three medical units at the time of present study, and patient admission is equally distributed among these units.

### Study enrollment

All clinically suspected cases of leptospirosis at THA were consecutively recruited for this study. The study was a component of a larger fever-surveillance study, and systematic assessment and investigations were carried out among all febrile patients admitted to the THA teaching unit from June 2012 to May 2013. Dengue, typhus, and leptospirosis were the primary focuses, and after excluding all other causes of fever, all remaining undifferentiated fever patients were assessed clinically for possible cases of leptospirosis. We previously showed that such a case definition may have low sensitivity—especially to detect anicteric leptospirosis and mild cases [11]; therefore, we did not use the surveillance case definition proposed by the WHO and adopted by the epidemiology unit of Sri Lanka for possible cases. Rather, we used a less stringent case definition as proposed earlier to enhance inclusivity [21]. All clinically suspected cases were considered as "possible" cases of leptospirosis.

### Patient recruitment and data collection

During the study period, the first author, a physician in the THA teaching unit assessed all febrile patients daily to consider each patient as a "possible" case for enrollment. All febrile cases were assessed using a symptom and sign checklist given upon admission, at which time a serum sample was also collected by a registered nurse. Data were collected using an interviewer-administered questionnaire. A standardized protocol was used to systematically record clinical and laboratory data. All patients were assessed daily while hospitalized. From day 1 of hospital admission, routine measurements included complete blood counts, serum creatinine, blood urea nitrogen, transaminases, alkaline phosphatase, bilirubin level, serum electrolytes, and urine biochemistry. In addition, a routine X-ray was done for all patients who exhibited at least one sign of respiratory involvement. All patients with probable cardiac involvement had electrocardiograms performed. All patients were given an appointment for follow-up assessment, and the examinations and routine investigations were carried out during the follow-up visit at 14 days post-discharge from the hospital. All samples were stored in -80˚C until further analysis.

All cases were confirmed according to the WHO Collaborating Leptospirosis Reference Laboratory in France using the microscopic agglutination test (MAT) with a broad panel of serovars (Table 1). We also used a lateral flow immunoassay (Immunemed *Leptospira*, Korea) as a bedside/point-of-care diagnostic test. (Dahanayaka *et al.*, 2017) Final diagnosis was established based on a combination of results from these tests. In MAT titre, single high titre>1/ 400, seroconversion of fourfold increase were considered as disease confirmation. Positive

**Table 1. Serovar used in the MAT panel.**

| Serial | Species | Serogroup | Serovar | Strain |
|---|---|---|---|---|
| 1 | *L. interrogans* | Australis | Australis | Ballico |
| 2 | *L. interrogans* | Autumnalis | Autumnalis | Akiyami A |
| 3 | *L. interrogans* | Bataviae | Bataviae | Van Tienen |
| 4 | *L. interrogans* | Canicola | Canicola | Hond Utrecht IV |
| 5 | *L. borgpetersenii* | Ballum | Castellonis | Castellon 3 |
| 6 | *L. kirschneri* | Cynopteri | Cynopteri | 3522 C |
| 7 | *L. kirschneri* | Grippotyphosa | Grippotyphosa | Moskva V |
| 8 | *L. interrogans* | Sejroe | Hardjobovis | Sponselee |
| 9 | *L. interrogans* | Hebdomadis | Hebdomadis | Hebdomadis |
| 10 | *L. interrogans* | Icterohaemorrhagiae | Copenhageni | Wijnberg |
| 11 | *L. noguchii* | Panama | Panama | CZ 214 K |
| 12 | *L. biflexa* | Semaranga | Patoc | Patoc 1 |
| 13 | *L. interrogans* | Pomona | Pomona | Pomona |
| 14 | *L. interrogans* | Pyrogenes | Pyrogenes | Salinem |
| 15 | *L. borgpetersenii* | Sejroë | Sejroë | M 84 |
| 16 | *L. borgpetersenii* | Tarassovi | Tarassovi | Mitis Johnson |
| 17 | *L. interrogans* | Icterohaemorrhagiae | Icterohaemorrhagiae | Verdun |
| 18 | *L. weilii* | Celledoni | ND | 2011/01963 |
| 19 | *L. interrogans* | Djasiman | Djasiman | Djasiman |
| 20 | *L. borgpetersenii* | Mini | ND | 2008/01925 |
| 21 | *L. weilii* | Sarmin | Sarmin | Sarmin |
| 22 | *L. santarosai* | Shermani | Shermani | 1342 K |
| 23 | *L. borgpetersenii* | Javanica | Javanica | Poi |
| 24 | *L. noguchii* | Louisiana | Louisiana | LUC1945 |

MAT titre without falling in to those categories or positive LFIA was considered as probable cases. All other clinically suspected cases were considered as possible cases.

## Multiplex cytokine assay

Multiplex cytokine assay was performed at Scripps Research Institute, La Jolla, Sand Diego. A magnetic bead–based multiplex cytokine kit (Bio-Plex Pro Human Cytokine, [Bio-Rad, Hercules, CA) was used to detect 17 cytokines: IL-1β, IL-2, IL-4, IL-5, IL-6, IL-7, IL-8, IL-10, IL-12, IL-13, IL-17, Interferon-γ (IFN-γ), granulocyte colony-stimulating factor (G-CSF), granulocyte-macrophage colony-stimulating factor (GM-CSF), monocyte chemoattractant protein (MCP-1), macrophage inflammatory protein (MIP-1b), and tumor necrosis factor (TNF-α). Briefly, antigens supplied with the kit were used to generate a standard eight-point dilution series. Samples were diluted 1:100 using the Bio-Plex standard diluent. The coupled beads (50 μL) followed by standard or test sample were added to the wells of a pre-wetted 96-well microtiter plate. After the recommended washing, plates were incubated for 30 min in a shaker (850 rpm) at 25°C. Detection antibody mixture (25 μL) was added, and the samples were incubated on ice for 30 min and then washed. Finally, 50 μL of streptavidin-phycoerythrinwas added, and after the recommended incubation [on ice for 10 min] and washing cycles, the beads were resuspended in 125 μL assay buffer and analyzed using the BioPlex suspension array system (Bio-Rad Laboratories) and Bio-Plex manager software. A curve fit was applied to each standard curve according to the manufacturer's manual, and sample concentrations were interpolated from the standard curves. Each sample was assessed in triplicate, and the mean cytokine concentration was taken as the final value.

Levels of individual cytokines were statistically aggregated using principal component analysis with Varimax rotation and Kaiser Normalization. This was done to understand the complex relationship between cytokines and the clinical outcomes.

Written informed consent was obtained from all eligible patients prior to enrollment in the study. Ethics clearance for this study was obtained from the Ethics Review Committee of the Faculty of Medicine and Allied Sciences, Rajarata University of Sri Lanka.

# Results

## Case confirmation

From June 2012 to May 2013, 142 clinically suspected leptospirosis patients were recruited to the study. The cohort included 121 males (85%) and 21 females (15%). Median age of the patients was 41 years (interquartile range 32–52 years) with approximately equal number of men and women. Paired serum samples were available for analysis among 79 (55.6%) patients. Of the study sample, 36 men, and 11 women were confirmed to have leptospirosis (Fig 1). In addition, 29 cases were labeled as "probable" based on the pre-set case definitions used. The rest (66) were considered as possible cases.

## Temporal distribution of cases

There was a clear peak of leptospirosis cases from December to March, which was compatible with the case notifications received by the regional epidemiologist for leptospirosis from Anuradhapura district (Fig 2).

## Exposure history

Of the 76 cases (confirmed and probable), an exposure history was available for 74. The most common exposure type was working in rice paddy fields (n = 63, 85%), proximity to marshy land (n = 32, 43%), having contact with stagnant water (n = 19, 26%), and having contact with surface waters (n = 16, 22%). All confirmed cases were able to provide a typical history of one of these types of, whereas 3 probable cases and 12 possible cases were unable to recall any likely leptospiral exposure.

## Clinical profile of patients

Median duration of fever on admission was 4 days (IQ range 2–5 days), with no difference among the confirmed, probable, and possible cases. Assessment of symptoms and signs on admission revealed that myalgia and arthralgia were the commonest clinical presentation associated with fever (Table 2). Classical features of leptospirosis, such as conjunctival suffusion and icterus, were present in ~25% of study participants. There was no significant difference (Chi square test, p < .05) between possible probable or confirmed cases and even after amalgamating confirmed and probable cases, results were same.

## Clinical evolution of leptospirosis among the cohort

All hospitalized study participants were clinically assessed daily by one of the investigators. For all study participants, routine laboratory tests were carried out daily, including white cell and neutrophil counts. Total white cell count, neutrophil count and platelet count was not significantly differ between confirmed/probable and possible cases (S1 Dataset). Further analysis was done only for confirmed and probable cases (Fig 3). Thrombocytopenia was common at the time of hospital admission. Total white cell and neutrophil counts dropped steadily until day 8, whereas the platelet count remained below the normal range throughout. Total white cell,

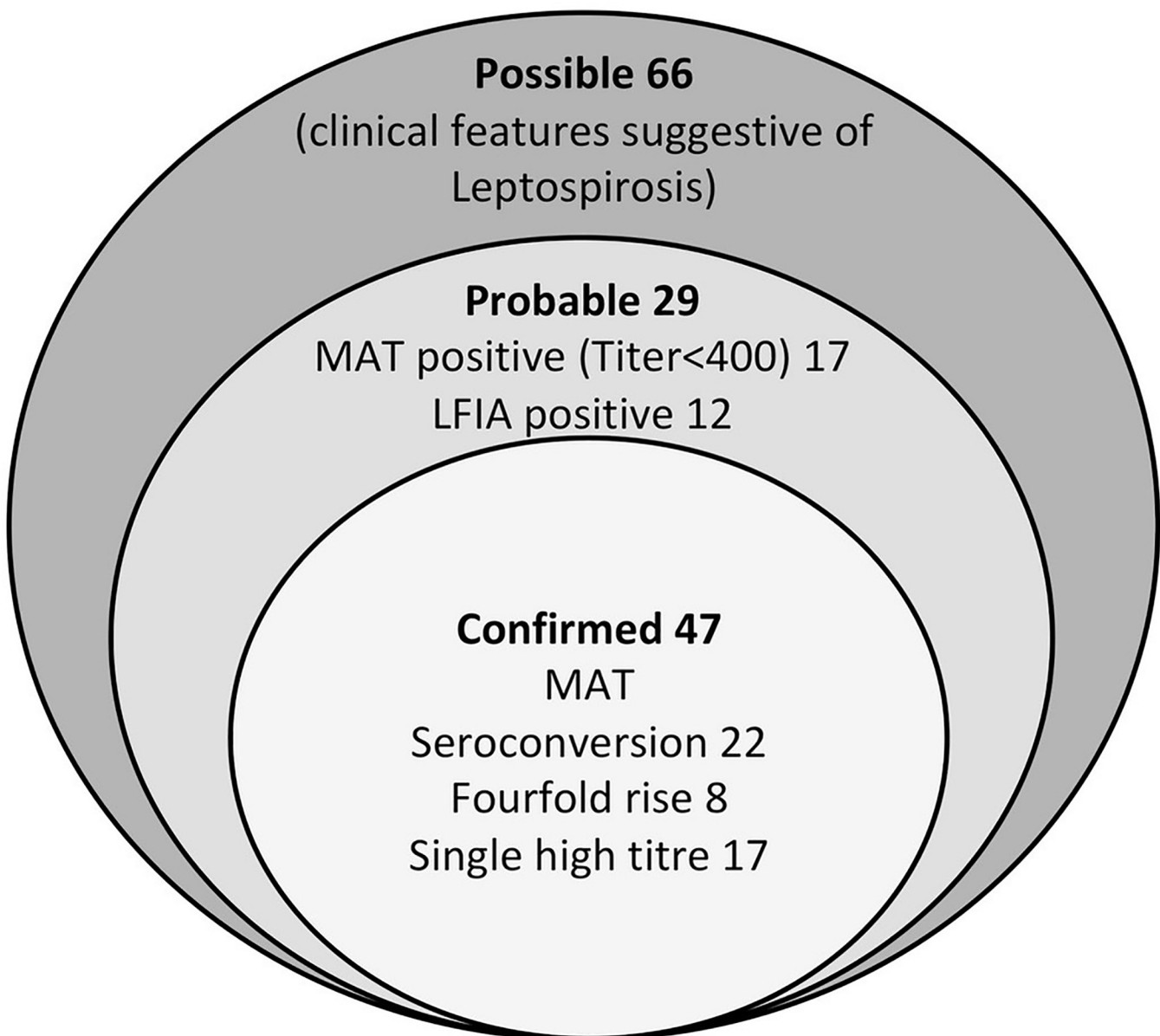

**Fig 1. Leptospirosis case confirmation among febrile patients admitted to the professorial medical unit, teaching hospital Anuradhapura from June 2012 to May 2013.**

neutrophil, and platelet counts returned to normal at 2 weeks post-discharge from the hospital among all cases. A slight increase of packed cell volume (PCV) on day 2–4 was observed among all confirmed cases (S1 Dataset). Of the 47 confirmed cases, 13 (27.6%) had PCV increase more than 20% of the baseline value.

## Complications and sequelae

Clinically significant hemorrhage was not observed in confirmed leptospirosis patients, despite thrombocytopenia. Acute kidney injury, defined as an increase in serum creatinine of >1.5-fold from baseline, was observed in 31 possible, probable and confirmed patients (22%).

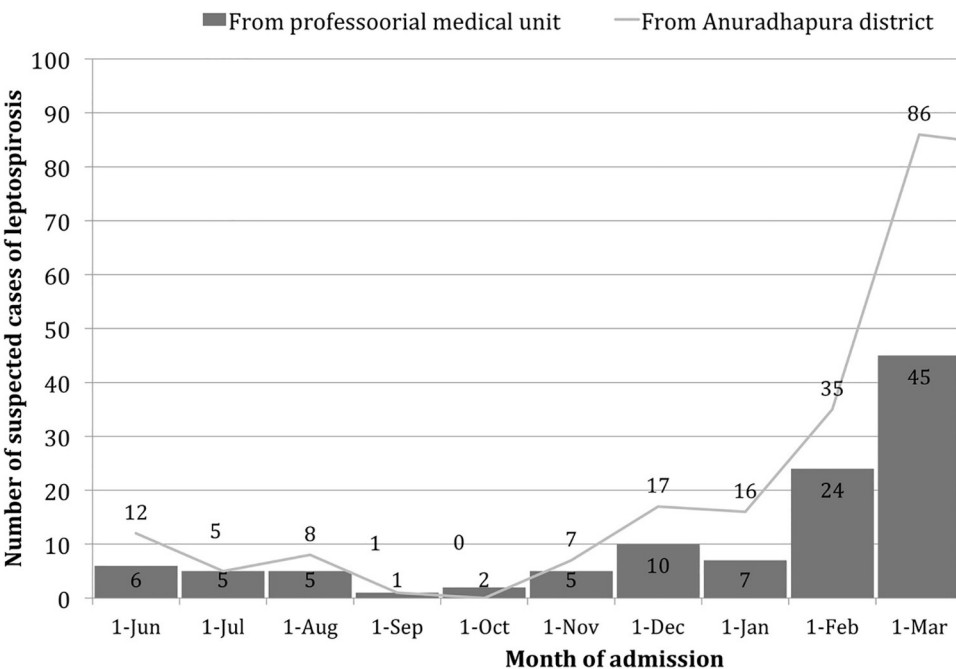

**Fig 2. Comparison of trends of leptospirosis cases admitted to the professorial medical unit, teaching hospital Anuradhapura from June 2012 to May 2013 and routinely reported data from Anuradhapura district, Sri Lanka during the same period.**

Among these, 15, 10, and 6 patients had risk, injury, or kidney failure, respectively, according to the RIFLE classification [22]. Confirmed/probable patients were having significantly higher percentage (31.6% vs 10.6%) of AKI compared to possible cases. Only AKI was significantly different between confirmed/probable vs possible cases (Table 3).

**Table 2. Clinical features of leptospirosis presented by patients admitted to the teaching hospital Anuradhapura, Sri Lanka from June 2012 to May 2013.**

| Clinical feature | Confirmed n % | | Probable % | | Possible n % | | Total n % | |
|---|---|---|---|---|---|---|---|---|
| Myalgia | 46 | 97.9 | 22 | 75.9 | 60 | 90.9 | 128 | 90.1 |
| Arthralgia | 42 | 89.4 | 25 | 86.2 | 59 | 89.4 | 126 | 88.7 |
| Retroorbital pain | 23 | 48.9 | 11 | 37.9 | 22 | 33.3 | 56 | 39.4 |
| Abdominal pain | 24 | 51.1 | 11 | 37.9 | 20 | 30.3 | 55 | 38.7 |
| Vomiting | 24 | 51.1 | 8 | 27.6 | 22 | 33.3 | 54 | 38.0 |
| Calf muscle tenderness | 16 | 34.0 | 6 | 20.7 | 23 | 34.8 | 45 | 31.7 |
| Conjunct injection | 18 | 38.3 | 6 | 20.7 | 14 | 21.2 | 38 | 26.8 |
| Hepatomegaly | 17 | 36.2 | 8 | 27.6 | 10 | 15.2 | 35 | 24.6 |
| Reduced urine output | 14 | 29.8 | 9 | 31.0 | 9 | 13.6 | 32 | 22.5 |
| Thigh muscle tenderness | 10 | 21.3 | 5 | 17.2 | 17 | 25.8 | 32 | 22.5 |
| Chest pain | 9 | 19.1 | 5 | 17.2 | 16 | 24.2 | 30 | 21.1 |
| Icterus | 13 | 27.7 | 5 | 17.2 | 11 | 16.7 | 29 | 20.4 |
| Diarrhea | 12 | 25.5 | 3 | 10.3 | 12 | 18.2 | 27 | 19.0 |
| RH tenderness | 10 | 21.3 | 2 | 6.9 | 6 | 9.1 | 18 | 12.7 |
| Epigastric tenderness | 4 | 8.5 | 4 | 13.8 | 2 | 3.0 | 10 | 7.0 |
| Palpitation | 3 | 6.4 | 2 | 6.9 | 4 | 6.1 | 9 | 6.3 |
| Flushed appearance | 2 | 4.3 | 1 | 3.4 | 6 | 9.1 | 9 | 6.3 |

None of these differences were significant (Chi square test, p>.05).

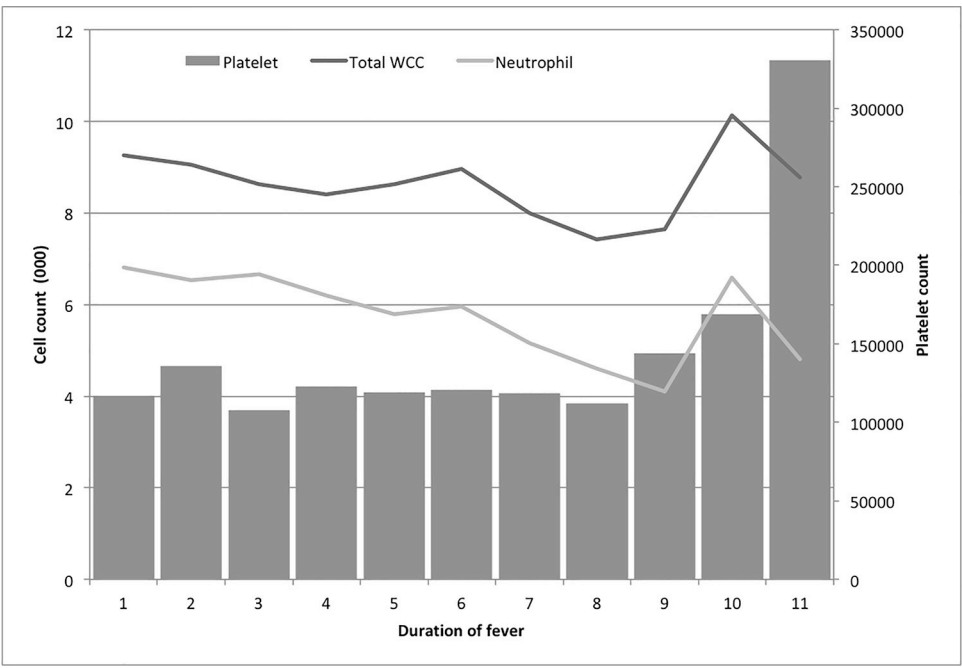

**Fig 3. Changes in total white cell, neutrophil, and platelet counts during the course of illness among probable and confirmed cases of leptospirosis admitted to the professorial medical unit, teaching hospital Anuradhapura, Sri Lanka from June 2012 to May 2013.**

The duration of hospitalization ranged from 1 to 12 days with a median of 3 days. During the study period, one death occurred (case fatality rate, 0.7%). This was a 51-year-old male part-time farmer whose fever had been present for 6 days before admission and he died on the 10th day of hospitalization from a syndrome that included oliguric renal failure, pulmonary hemorrhage and refractory shock. Serum creatinine was 140 μmol/L (normal range 90–120 μmol/L) on admission. This patient had a MAT titer of 1/200 and a positive Leptospira IgM in LFIA.

**Table 3. Complications observed among patients with leptospirosis admitted to the professorial medical unit from June 2012 to May 2013, and routinely reported data from the Anuradhapura district during the same period.**

|  | Confirmed (47) | | Probable (29) | | Possible (66) | | Total | |
|---|---|---|---|---|---|---|---|---|
|  | n | % | n | % | n | % | n | % |
| **Liver involvement** | | | | | | | | |
| SGPT>60 ul | 17 | 36.2 | 4 | 13.79 | 23 | 34.9 | 44 | 31.0 |
| Bilirubin (direct)>7 mmol/L | 4 | 8.5 | 1 | 3.45 | 5 | 7.6 | 10 | 7.1 |
| Acute kidney injury | 19 | 42.5 | 7 | 36.8 | 11 | 14.9 | 31 | 21.8 |
| Hypotension or shock | | | | | | | | |
| SBP<90mmHg | 15 | 31.9 | 4 | 13.79 | 11 | 16.7 | 30 | 21.1 |
| Required inotropic support | 10 | 21.3 | 2 | 6.90 | 5 | 7.6 | 17 | 12.0 |
| Lung involvement | | | | | | | | |
| Saturation drop <92% | 1 | 2.1 | 1 | 3.45 | 6 | 9.1 | 8 | 5.6 |
| Diffuse alveolar shadowing | 4 | 8.5 | 3 | 10.3 | 3 | 4.6 | 10 | 7.0 |
| Myocarditis | 1 | 2.1 | 0 | 0.0 | 3 | 4.6 | 4 | 2.8 |

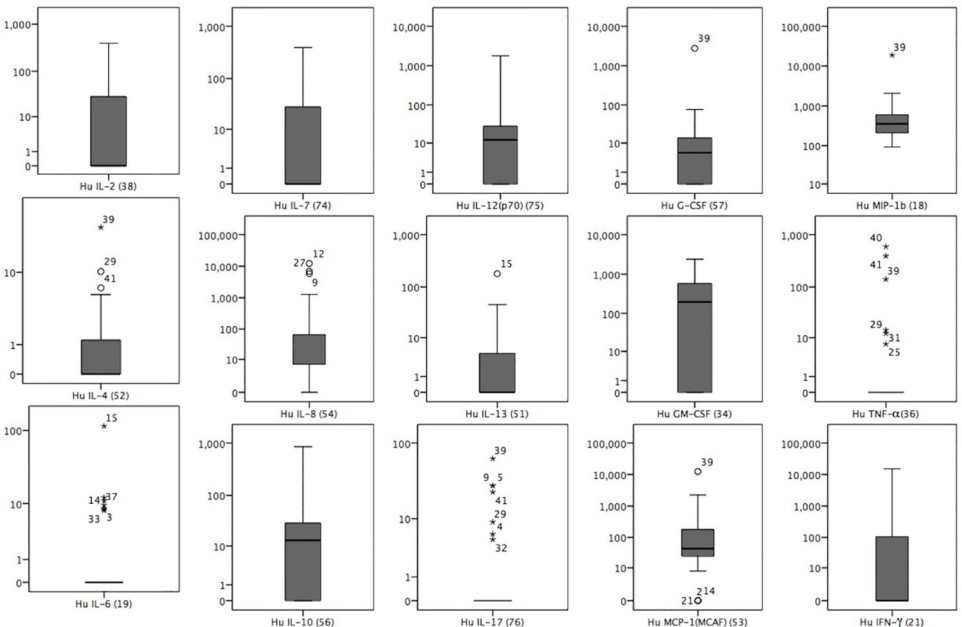

**Fig 4. Distribution of cytokines during the acute phase of illness among probable and confirmed cases of leptospirosis admitted to the professorial medical unit, teaching hospital Anuradhapura from June 2012 to May 2013.**

## Cytokine analysis of study participants

We prospectively measured serum cytokines in patients suspected to have leptospirosis, with the long-term goal of identifying potential a biosignature of acute leptospirosis. Though acute sample cytokines were done for 43 confirmed/ probable cases, complete clinical and diagnostic data were available from 41patients. The curve fit statistics of the cytokine analysis shows that the generated data are not deviated from the standard curve, confirming the validity of cytokine estimates. The metadata include 30 confirmed and 11 probable cases. Human (Hu) MIP-1b was detected in all samples, and Hu MCP-1 (also called MCAF), Hu IL-8, Hu IL-6, and Hu GM-CSF were detected in more than 75% of the samples. Hu IL-5 and IL-1β were detected in only 2 and 4 samples, respectively, and hence were excluded from the cytokine analysis of the acute samples (Fig 4).

We compared the distribution of cytokines among those patients who had complications with those who lacked complications (Table 4). Of the 41 patients studied, 17 (42%) had one or more complications. Impairment of renal function (serum creatinine >1 mg/dL for females, >1.2 mg/dL for males) was observed for 16 patients (39%). Eleven patients (27%) had hypotension (systolic blood pressure <90 mmHg, and diastolic blood pressure <60 mmHg), and 6 patients (15%) required treatment with inotropic agents. Six (15%) patients had pulmonary involvement with abnormal chest X-rays, and 3 patients required intubation and ventilator support.

Hu TNF-α and IL-1β were detected only in patients without complications. Hu MIP-1b was detected in all 41 patients, and the levels were significantly higher among patients with complications. While Hu IL-8 appeared to be **elevated** among patients with complications, the difference was not statistically significant from uncomplicated cases.

Of the 41 leptospirosis cases with acute sample data, only 14 were followed up as outpatients (8 confirmed and 6 probable), and samples from convalescent patients were available with at

**Table 4. Distribution of cytokines during the acute phase of illness among confirmed and probable leptospirosis patients with or without complications.**

|  | Complications | | | | p value |
|---|---|---|---|---|---|
|  | Yes (n = 17) | | No(n = 24) | | |
|  | Range | Median | Range | Median | |
| Hu IL-2 | 0–45.4 | 0.0 | 0–394.5 | 1.0 | 0.117 |
| Hu IL-4 | 0–3.5 | 0.0 | 0–30.7 | 0.0 | 0.898 |
| Hu IL-5 | 0–90.0 | 0.0 | 0–10.0 | 0.0 | 0.777 |
| Hu IL-6 | 0–174.7 | 30.2 | 0–99499.4 | 19.5 | 0.285 |
| Hu IL-7 | 0–114.6 | 0.0 | 0–12.2 | 0.0 | 0.936 |
| Hu IL-8 | 0–12196.1 | 50.5 | 0–6872.1 | 13.1 | 0.067 |
| Hu IL-10 | 0–712.0 | 12.2 | 0–846.8 | 13.7 | 0.989 |
| Hu IL-12 | 0–304.5 | 0.0 | 0–1773.5 | 13.5 | 0.34 |
| Hu IL-13 | 0–180.5 | 0.0 | 0–45.8 | 0.0 | 0.869 |
| Hu IL-17 | 0–28.0 | 0.0 | 0–63.0 | 0.0 | 0.904 |
| Hu G-CSF | 0–14.8 | 5.7 | 0–2759.0 | 5.1 | 0.58 |
| Hu GM-CSF | 0–927.2 | 215.2 | 0–2381.2 | 144.0 | 0.778 |
| Hu IFN-γ | 0–600.3 | 0.0 | 0–15066.5 | 0.0 | 0.946 |
| Hu MCP-1 (MCAF) | 0–481.9 | 61.8 | 0–12375.9 | 42.6 | 0.853 |
| Hu MIP-1b | 207–1436.6 | 497.5 | 92–18771.9 | 300.2 | 0.028 |
| Hu TNF-α | 0–0.0 | 0.0 | 0–592.7 | 0.0 | 0.028 |
| IL-1β | 0–0.0 | 0.0 | 0–1808 | 0.0 | 0.081 |

The Mann Whitney U test was used for data analysis. None of the differences were significant.

least 10 days from the acute phase to convalescence. Of the 14 cases for which paired samples available, 7 (50%) had one or more complications. The distribution of serum cytokine level during the convalescence period was not significantly different between patients with or without complications (Table 5). Analysis of individual cytokines and complications shows that different cytokines are associated with different outcomes (Table 6).

During the convalescence period, all tested serum cytokine levels were lower compared to the acute sample, except for IL-8 (Fig 5). Changes in serum cytokine levels from acute-stage samples to convalescence-stage samples were tested using the paired-sample t test. None of the cytokines showed significant differences in paired samples between patients with and without complications.

The principal component analysis showed a four factor solution for cytokines which explained 87.6% of the variation (Bartlett'e test of sphericity, chi-square 1298, p < .001). As shown in Table 7, the first component included Hu G-CSF, Hu IL-6, Hu MIP-1b, Hu MCP-1 (MCAF), Hu IL-4 and IL-1b. IL-8 loaded as a separate component on its own (Table 6). IL-5, IL-7 and IL-13 were in a separate group with rest in another group. The observed groups were not the typical proinflammatory or anti-inflammatory groups as described in the literature. The cytokines group 1 was positively associated with increased serum creatinine levels (MVU 136, p .087) and the group 4 with Thrombocytopenia.

## Discussion

Predicting the evolution of acute leptospirosis to severe, complicated disease remains a challenge owing to wide variation of clinical disease, complex disease transmission pattern, and variability of local mammalian reservoir hosts, local ecology, climate, host animals and virulence of local, geographically dominant infecting *Leptospira*. Where leptospirosis is endemic—usually in

**Table 5. Distribution of cytokines during the convalescence phase among confirmed and probable leptospirosis patients with or without complications.**

| | Complications | | | | p value |
| | Yes (n = 7) | | No (n = 7) | | |
| | Range | Median | Range | Median | |
|---|---|---|---|---|---|
| Hu IL-2 | 0–45.9 | 0 | 0–63.4 | 14.6 | 0.535 |
| Hu IL-4 | 0–5.1 | 0 | 0–12.7 | 0 | 1.000 |
| Hu IL-5 | 0–10.7 | 0 | 0–0 | 0 | 0.710 |
| Hu IL-6 | 0–2265.5 | 0 | 0–20088.4 | 7.1 | 0.805 |
| Hu IL-7 | 0–24.4 | 0 | 0–0 | 0 | 0.710 |
| Hu IL-8 | 0–7736.2 | 53 | 0–2630.7 | 34.7 | 0.902 |
| Hu IL-10 | 0–260.2 | 0 | 0–44.5 | 0 | 0.620 |
| Hu IL-12 | 0–251.8 | 19.7 | 0–48.9 | 0 | 0.318 |
| Hu IL-13 | 0–50.1 | 0 | 0–4.6 | 0 | 0.383 |
| Hu IL-17 | 0–25.8 | 0 | 0–0 | 0 | 0.383 |
| Hu G-CSF | 0–28.8 | 0 | 0–253 | 0 | 1.000 |
| Hu GM-CSF | 0–781 | 518.2 | 126.7–775.6 | 290.5 | 1.000 |
| Hu IFN-γ | 0–572.9 | 46.4 | 0–876.8 | 0 | 0.535 |
| Hu MCP-1 (MCAF) | 21–2631.7 | 36.7 | 0–3387.2 | 28.3 | 0.620 |
| Hu MIP-1b | 105.2–6199.4 | 387.6 | 0–451.7 | 299.2 | 0.259 |
| Hu TNF-α | 0–51 | 0 | 0–424.4 | 0 | 1.000 |
| IL-1β | 0–58.9 | 0 | 0–880.1 | 0 | 0.710 |

The Mann Whitney U test was used for data analysis.

resource-limited settings—routine diagnostic facilities are not typically available. Hence reliable case reporting is generally not available leading to an underestimation of local as well as global disease burden [23]. Disease burden estimates based on country-specific statistics are not precise because reported data are biased in many ways, particularly based on the identification of typical and severe case presentation for which diagnostic testing was sought.

**Table 6. Acute phase cytokine levels and selected outcomes among confirmed and probable leptospirosis patients.**

| Complication | Associated Cytokine | Significance |
|---|---|---|
| Thrombocytopenia (Platelet <150,000) | IL-8 | p-0.02 MVU-100.000 |
| Cardiac involvement requiring Inotropes | MIP-1b | p-0.01 MVU-39.000 |
| Billirubin (direct)>7 mmol/l | IL-13 | p-0.035 MVU -11.000 |
| High serum creatinine (Serum creatinine>100mg/dl for females/120mg/dl per males) | GM-CSF | p-0.03 MVU-123.000 |
| Oxygen given | MIP-1b | p-0.06 MVU-55.000 |
| Lung involvement | IL-5 | p-.045 MVU-57.000 |
| Saturation drop | IL-6 | p-0.07 MVU-10.000 |
| Hypotension | MIP-1b | p-0.08 MVU-107.000 |
| | IL-8 | p-0.09 MVU-109.000 |

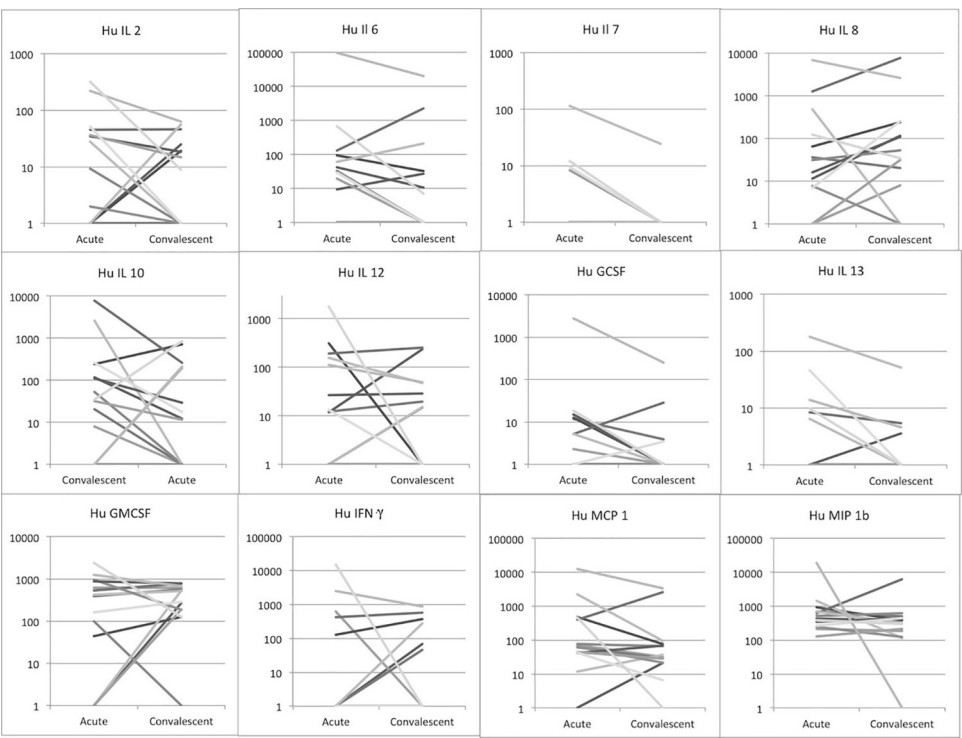

**Fig 5. Distribution of serum cytokine levels in samples from acute and convalescent patients among probable and confirmed cases of leptospirosis from the professorial medical unit, teaching hospital Anuradhapura from June 2012 to May 2013.**

Despite the prospective nature of the present study, in which sensitive criteria for suspecting cases was adopted and rigorous case definition was employed, we were not able to identify a comprehensive cytokine biosignature of acute or complicated leptospirosis. Routine laboratory testing yield results consistent with previous reports, for example more prominent changes with the same pattern of complete blood counts and serum chemistries, as reported by de Silva et al. (2014) from Colombo, Sri Lanka. This may be attributable to the wide clinical spectrum among patients in our study compared with the study of Silva et al. [24]. Here, we observed that leukocytosis not a feature of leptospirosis in our study region, even though this is expected for bacterial infections. However leukopenia usually differentiates leptospirosis from dengue, which typical is associated with low white blood cell counts. Thrombocytopenia was observed in more than half of our study cohort, which might complicate the clinical diagnosis because of potential confusion with dengue, in places such as Sri Lanka where both infections are transmitted. Hematoconcentration might be a helpful differentiator, a laboratory maker in dengue. However, out data suggested that among confirmed leptospirosis patients, around one fourth were having marked variation in PCV during hospital stay, which may complicate the diagnosis furthermore.

Patterns of leptospirosis complications and clinical features observed in the present study differ from those previously published during a post-flood outbreak of leptospirosis [12]. We observed a high level of renal involvement (55% compared with 22%), but myocarditis was less common (2% compared with 16% among confirmed cases). In addition, we observed that nearly one-third of the confirmed cases developed hypotension with 20% requiring treatment with inotropic agents, which was not observed in our previous studies [11,12]. Observing these markedly different clinical features and outcomes strengthens the hypothesis that endemic

**Table 7. Factor analysis of cytokine response.**

| Rotated Component Matrix[a] | | | | |
|---|---|---|---|---|
| | **Component** | | | |
| | **1** | **2** | **3** | **4** |
| Hu G-CSF | 0.989 | | | |
| Hu IL-6 | 0.988 | | | |
| Hu MIP-1b | 0.981 | | | |
| Hu MCP-1(MCAF) | 0.974 | | | |
| Hu IL-4 | 0.907 | | | |
| IL-1b | 0.753 | | | |
| Hu IFN-g | | 0.978 | | |
| Hu TNF-a | | 0.967 | | |
| Hu IL-12(p70) | | 0.876 | | |
| Hu IL-2 | 0.323 | 0.837 | | |
| Hu GM-CSF | | 0.829 | | |
| Hu IL-10 | | 0.694 | | |
| Hu IL-5 | | | 0.989 | |
| Hu IL-7 | | | 0.987 | |
| Hu IL-13 | | | 0.984 | |
| Hu IL-8 | | | | 0.993 |

Extraction Method: Principal Component Analysis.

Rotation Method: Varimax with Kaiser Normalization.

[a] Rotation converged in 4 iterations.

and epidemic leptospirosis may differ even in the same regional setting, and this is probably attributable to different infecting *Leptospira* species or serovars present there [12]. This is a reasonable working hypothesis in countries like Sri Lanka where at least five different species of *Leptospira* that can cause human leptospirosis are currently circulating, with more than 40 identified serovars [25].

The presently reported cytokine profiles in severe and mild leptospirosis cases, with a subset of such cases including both acute and convalescence phases, resulted in more questions than answers. "Cytokine storm" has been previously reported as a marker of disease severity in leptospirosis [26]. Our prospective study did not confirm such results, but instead results were more complex, with elevated IL-6 and IL-8 among patients with complications. These findings are compatible with previous cross-sectional studies. In contrast we could not confirm a previously reported difference in mean IL-10 levels among severe and mild cases [15,17,27]. IL-10 is known to regulate inflammation by down-regulating monocyte-derived TNF-α and IL-1 [28]. TNF-α was undetectable in most cases, but if detected it was only in patients without complications. Acute-phase MIP-1b level ranged from 92 to 18772 units for patients without complications and from 207 to 1437 for patients with complications. A previous study reported higher levels of MIP-1b (11–2442 units) in severe cases and lower levels (0–765 units) in mild cases [27]. These findings are contradictory but could be attributable to differences in host response to different serovars that might be present in the two study settings. In acute and convalescence phases, we observed marked reduction of almost all cytokines indicated disease resolution. Papa et al. (2015) reported a gradual reduction of IL-6 and IL-8 within the first 5 days of illness and an increase thereafter [27], which differs from what we previously observed [27]. During the convalescence phase, however, the levels of both IL-6 and IL-8 were lower, and there was no significant change in their levels throughout convalescence.

In this study, we were unable to identify a biosignature of leptospirosis using a large panel of cytokine measurements. As reported in several other studies, our study clearly shows the inadequacy of routinely reported data for disease burden estimates even in a known hyperendemic region. The district of Anuradhapura has six other hospitals with more than 100 beds, and another 32 small hospitals with inpatient facilities. Throughout the study period, however, we discovered that more than half of the number of leptospirosis patients in the district were reported by one of the various university teaching units. These data clearly suggest that more than half of the suspected inpatients did not diagnose leptospirosis. In addition, patients seeking outpatient care–which is likely the majority of cases—were not included in these estimates, which biases both reported and understanding the basis of severe leptospirosis disease and complications. This underreporting could be due to a lack of clinical suspicion, as we observed previously [11], lack of diagnostic facilities owing to problems with reporting, or these factors in combination. Current data indicate that approximately 4,000 cases of leptospirosis are reported annually to the central epidemiology unit of the Sri Lanka Ministry of Health. However, the actual number may exceed 10,000 based on observations reported in the present and previous studies [12]. Systematic, population-based studies of patients with undifferentiated fever with efficient diagnostics are to support better public health policies to control this disease on local, regional, national and international levels.

The results of this study have to be interpreted within the study limitations. The analysis is based on confirmed probable and possible cases. On clinical grounds, we have not seen marked differences in probable and possible cases. In a setting where other tropical diseases such as hantavirus infection are common, these probable cases may affect the results. However, these observations, especially the lack of association, needs to be documented in the scientific literature to avoid publication bias. For a better interpretation of results, all data used are provided as supporting information file (S1 Dataset) so that other researchers could use the raw data for further analysis.

## Supporting information

**S1 Dataset. The clinical, biochemical and cytokine data of patients.**
(SAV)

## Acknowledgments

We acknowledge Mathieu Picardeau and National Reference Center and WHO Collaborating Center for Leptospirosis, Institut Pasteur, Paris, France for MAT testing.

## Author Contributions

**Conceptualization:** Niroshana J. Dahanayaka, Suneth B. Agampodi, Michael Matthias.

**Data curation:** Niroshana J. Dahanayaka, Suneth B. Agampodi, Rukman Rajapakse, Kosala Ranathunga.

**Formal analysis:** Suneth B. Agampodi, Indika Seneviratna, Janith Warnasekara, Michael Matthias.

**Funding acquisition:** Niroshana J. Dahanayaka, Suneth B. Agampodi, Joseph M. Vinetz.

**Investigation:** Niroshana J. Dahanayaka, Suneth B. Agampodi, Indika Seneviratna, Rukman Rajapakse, Kosala Ranathunga, Michael Matthias.

**Methodology:** Niroshana J. Dahanayaka, Suneth B. Agampodi, Indika Seneviratna, Janith Warnasekara, Michael Matthias, Joseph M. Vinetz.

**Project administration:** Niroshana J. Dahanayaka, Suneth B. Agampodi.

**Resources:** Suneth B. Agampodi, Joseph M. Vinetz.

**Software:** Suneth B. Agampodi.

**Supervision:** Suneth B. Agampodi, Michael Matthias, Joseph M. Vinetz.

**Validation:** Suneth B. Agampodi.

**Visualization:** Suneth B. Agampodi.

**Writing – original draft:** Niroshana J. Dahanayaka, Suneth B. Agampodi, Indika Seneviratna, Janith Warnasekara.

**Writing – review & editing:** Suneth B. Agampodi, Michael Matthias, Joseph M. Vinetz.

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
