## [Decision Letter · Decision Letter 0]

2 Nov 2021

PONE-D-21-30348Clinical spectrum of endemic leptospirosis in relation to cytokine responsePLOS ONE

Dear Dr. Agampodi,

Thank you for submitting your manuscript to PLOS ONE. After careful consideration, we feel that it has merit but does not fully meet PLOS ONE’s publication criteria as it currently stands. Therefore, we invite you to submit a revised version of the manuscript that addresses the points raised during the review process. Your manuscript has been reviewed by an expert in your field.  Based the reviewer comments, a  major revision is needed before a decision can be make.

Please submit your revised manuscript on Dec 17, 2021. If you will need more time than this to complete your revisions, please reply to this message or contact the journal office at plosone@plos.org. Please include the following items when submitting your revised manuscript:A rebuttal letter that responds to each point raised by the academic editor and reviewer(s). You should upload this letter as a separate file labeled 'Response to Reviewers'.A marked-up copy of your manuscript that highlights changes made to the original version. You should upload this as a separate file labeled 'Revised Manuscript with Track Changes'.An unmarked version of your revised paper without tracked changes. You should upload this as a separate file labeled 'Manuscript'.

We look forward to receiving your revised manuscript.

Kind regards,

Yung-Fu Chang

Academic Editor

PLOS ONE

Journal Requirements:

Reviewers' comments:

Reviewer's Responses to Questions

**Comments to the Author**

1. Is the manuscript technically sound, and do the data support the conclusions?

Reviewer #1: Yes

2. Has the statistical analysis been performed appropriately and rigorously? 

Reviewer #1: Yes

3. Have the authors made all data underlying the findings in their manuscript fully available?

Reviewer #1: Yes

4. Is the manuscript presented in an intelligible fashion and written in standard English?

Reviewer #1: Yes

5. Review Comments to the Author

Reviewer #1: The aim of the study was to describe clinical and laboratory features (including cytokine profile - biosignature) of leptospirosis patients in Sri Lanka.

First there is a sample problem. Of the 142 recruited patients, confirmed leptospirosis diagnosis was made only in 47. There were 95 probable or possible cases, based on clinical features (66) and doubtful lab results (29). These harms the analysis. Symptoms suggestive of leptospirosis can be confounded with many other febrile illnesses. Severe cases, more easily diagnosed, were the minority. Clinical features of confirmed, probable and possible cases were the same.

Symptoms suggestive of leptospirosis (calf muscle pain - 31.7%, conjunctival injection - 26.8%, flushed appearance - 6.3%, jaundice - 20.4%) were little frequent. General symptoms, like myalgia, headache, vomiting, etc.) were more frequent.

Second, little more than half of the sample had paired samples for serology. I know it is difficult to get follow-up samples, but it was a clinical study and patients should had been stimulated to come for a follow-up visit and sample collection, to improve the diagnosis yield.

Third, there is no difference between confirmed, probable and possible cases in clinical grounds. This is expected regarding differential diagnosis of leptospirosis in acute febrile illnesses context. So, there is no guarantee that patients with other diseases where included in the analysis, spoiling results.

Figure 3 is quite confusing. What does it means? What is the reaction between duration of fever and cell counts?

Table 3 - confirmed cases had more AKI and shock, but possible cases had more lung involvement and myocarditis. Could it be other disease? Did you considered the possibility of hantavirus?

Does an autopsy was performed on the dead patient? Any attempt to collect tissue for pathological examination and diagnosis?

Cytokine analysis to identify a biosignature of acute leptospirosis - only 41 confirmed (30) and probable cases (11) available for acute analysis. Cytokine profile of complicated cases differed from non complicated cases only on MIP-1b and TNF alpha, but no TNF was detected on complicated cases. This is awkward and unexpected. TNF alpha was expected to be detected in high levels in complicated cases with AKI, shock and lung involvement. By the same way, MIP-1b levels were significantly higher in non complicated cases. These can be observed for the other cytokines levels range. I would like to see the mean and s.d. of cytokines levels instead of the median. MIP-1b broader range on non complicated patients suggests it is not good for case severity discrimination.

Paired cytokine analysis was available in 14 cases (8 confirmed and 6 probable), a very low number os cases. Its impossible to infer a cytokine profile or biosignature with such a low number os patient analysed.

Maybe the choice of statistical method should be reviewed. It is very hard for a medical doctor to deal with obscure statistical techniques. Risk analysis, logistic regression and other more common statistics could be used. Despite this, it is clear to me that no cytokine profile could be inferred from these sample. I consider difficult or even impossible to obtain a cytokine signature for leptospirosis. As you said in discussion, there are many different kinds of leptospirosis, depending on serovar and host. It is a complex relationship. Leptospirosis immune pathogenesis is not yet well understood. We need to study how leptospira express proteins and produces tissue lesion and how host response developes after infection. Epidemiological and social factors may influence response, like nutritional status, previous diseases, etc.

As in bacterial sepsis, probably we will not be able to define a cytokine biosignature for severe leptospirosis.

More efforts must be concentrated in stimulating clinical suspicion and fast laboratory diagnosis, as well as in understanding disease mechanisms.

6. PLOS authors have the option to publish the peer review history of their article (what does this mean?). If published, this will include your full peer review and any attached files.

Reviewer #1: **Yes: **Decio Diament

---

## [Author Response · Author response to Decision Letter 0]

12 Nov 2021

The manuscript file was edited accordingly. 

 We note that you have included the phrase “data not shown” in your manuscript. Unfortunately, this does not meet our data sharing requirements. PLOS does not permit references to inaccessible data. We require that authors provide all relevant data within the paper, Supporting Information files, or in an acceptable, public repository. Please add a citation to support this phrase or upload the data that corresponds with these findings to a stable repository (such as Figshare or Dryad) and provide and URLs, DOIs, or accession numbers that may be used to access these data. Or, if the data are not a core part of the research being presented in your study, we ask that you remove the phrase that refers to these data.

We have edited the manuscript and included the full data set as a supporting information.

Reply to reviewers 

Reviewer #1: The aim of the study was to describe clinical and laboratory features (including cytokine profile - biosignature) of leptospirosis patients in Sri Lanka.

First there is a sample problem. Of the 142 recruited patients, confirmed leptospirosis diagnosis was made only in 47. There were 95 probable or possible cases, based on clinical features (66) and doubtful lab results (29). These harms the analysis. Symptoms suggestive of leptospirosis can be confounded with many other febrile illnesses. Severe cases, more easily diagnosed, were the minority. Clinical features of confirmed, probable and possible cases were the same.

Symptoms suggestive of leptospirosis (calf muscle pain - 31.7%, conjunctival injection - 26.8%, flushed appearance - 6.3%, jaundice - 20.4%) were little frequent. General symptoms, like myalgia, headache, vomiting, etc.) were more frequent.

Second, little more than half of the sample had paired samples for serology. I know it is difficult to get follow-up samples, but it was a clinical study and patients should had been stimulated to come for a follow-up visit and sample collection, to improve the diagnosis yield.

Third, there is no difference between confirmed, probable and possible cases in clinical grounds. This is expected regarding differential diagnosis of leptospirosis in acute febrile illnesses context. So, there is no guarantee that patients with other diseases where included in the analysis, spoiling results.

We humbly accept the point raised. These are common but major issues in clinical research in resource poor settings. We have included these as limitation of our study.

“The results of this study have to be interpreted within the study limitations. The analysis is based on confirmed probable and possible cases. On clinical grounds, we have not seen marked differences in probable and possible cases. In a setting where other tropical diseases such as hantavirus infection are common, these probable cases may affect the results. However, these observations, especially the lack of association, needs to be documented in the scientific literature to avoid publication bias. For a better interpretation of results, all data used are provided as supporting information file (S1 Dataset) so that other researchers could use the raw data for further analysis..” 

In addition, the observation that there was no clear difference in probable and possible cases is a major clinical dilemma clinicians are having. Unfortunately, these are not often published in literature, while only the “significant differences” are published leading to a publication bias.

Figure 3 is quite confusing. What does it means? What is the reaction between duration of fever and cell counts?

The figure shows the daily changes of cell counts over the course of illness. This is important for clinicians in these settings where they mostly use clinical features and basic investigations to understand the disease. 

Table 3 - confirmed cases had more AKI and shock, but possible cases had more lung involvement and myocarditis. Could it be other disease? 

This is actually quite common in leptospirosis. The lung involvement is more common with certain species of leptospirosis, while renal involvement is common in some other species. We have previously reported these microgeographical changes and carried out several studies to confirm this. We have included this point in the discussion.

“Patterns of leptospirosis complications and clinical features observed in the present study differ from those previously published during a post-flood outbreak of leptospirosis. We observed a high level of renal involvement (55 % compared with 22%), but myocarditis was less common (2% compared with 16 % among confirmed cases). In addition, we observed that nearly one-third of the confirmed cases developed hypotension with 20% requiring treatment with inotropic agents, which was not observed in our previous studies. Observing these markedly different clinical features and outcomes strengthens the hypothesis that endemic and epidemic leptospirosis may differ even in the same regional setting, and this is probably attributable to different infecting Leptospira species or serovars present there.”

Did you considered the possibility of hantavirus?

The problem of hantavirus infection is a great concern as suggested. Some of these cases could be hantavirus as shown by a recent study done in the same setting. We have added this as a limitation in the discussion. 

Does an autopsy was performed on the dead patient? Any attempt to collect tissue for pathological examination and diagnosis?

No. This could have been a major addition to the study if we could have done. But unfortunately no postmortem samples were taken. 

Cytokine analysis to identify a biosignature of acute leptospirosis - only 41 confirmed (30) and probable cases (11) available for acute analysis. Cytokine profile of complicated cases differed from non complicated cases only on MIP-1b and TNF alpha, but no TNF was detected on complicated cases. This is awkward and unexpected. TNF alpha was expected to be detected in high levels in complicated cases with AKI, shock and lung involvement. By the same way, MIP-1b levels were significantly higher in non complicated cases. These can be observed for the other cytokines levels range. I would like to see the mean and s.d. of cytokines levels instead of the median. MIP-1b broader range on non complicated patients suggests it is not good for case severity discrimination. Paired cytokine analysis was available in 14 cases (8 confirmed and 6 probable), a very low number os cases. Its impossible to infer a cytokine profile or biosignature with such a low number os patient analysed.

We agree with this point and the reason for reporting this study is also to have these data in the literature. We firmly believe that publication bias involving “no association” has heavily impacted clinical practice where the journal publication process systematically selecting only “positive associations”. Based on the reviewers comments, we have included all cytokine data with other clinical data as a supporting information file, so that it will be available for readers to download and interpret. As shown in the supplementary file data, the cytokine distribution is extremely skewed and showing the mean and SD will distort the results. This was the reason for using median. However, the supplementary file is provided now with all data. 

We have included following points in the discussion to discuss these issues.

“In this study, we were unable to identify a biosignature of leptospirosis using a large panel of cytokine measurements.”

Maybe the choice of statistical method should be reviewed. It is very hard for a medical doctor to deal with obscure statistical techniques. Risk analysis, logistic regression and other more common statistics could be used. 

The statistical test used in one of the most simple test (Mann Whitney U test). This is the test that can be used for highly skewed distribution. The suggested statistical tests are not useful for the intended analysis. The factor analysis is required because cytokine profiles could not be analyzed separately. (and the analysis was done by two medical doctors)

Despite this, it is clear to me that no cytokine profile could be inferred from these sample. I consider difficult or even impossible to obtain a cytokine signature for leptospirosis. As you said in discussion, there are many different kinds of leptospirosis, depending on serovar and host. It is a complex relationship. Leptospirosis immune pathogenesis is not yet well understood. We need to study how leptospira express proteins and produces tissue lesion and how host response developes after infection. Epidemiological and social factors may influence response, like nutritional status, previous diseases, etc.

As in bacterial sepsis, probably we will not be able to define a cytokine biosignature for severe leptospirosis.

More efforts must be concentrated in stimulating clinical suspicion and fast laboratory diagnosis, as well as in understanding disease mechanisms.

Agree fully with this comment. This was the reason for us to include the conclusion (highlighted) 

“Results of this study confirms that the knowledge on cytokine response in leptospirosis could be more complex than other similar tropical disease, and biosignatures that provide diagnostic and prognostic information for human leptospirosis remain to be discovered.”

We have included these points in the discussion. And as mentioned above It will be important to have these data as evidence to show that cytokine biosignature is not very useful. Thus the data are also included in the manuscript. 

“The results of this study have to be interpreted within the study limitations. The analysis is based on confirmed probable and possible cases. On clinical grounds, we have not seen marked differences in probable and possible cases. In a setting where other tropical diseases such as hantavirus infection are common, these probable cases may affect the results. However, these observations, especially the lack of association, needs to be documented in the scientific literature to avoid publication bias. For a better interpretation of results, all data used are provided as supporting information file (S1 Dataset) so that other researchers could use the raw data for further analysis..”

---

## [Decision Letter · Decision Letter 1]

23 Nov 2021

Clinical spectrum of endemic leptospirosis in relation to cytokine response

PONE-D-21-30348R1

Dear Dr. Agampodi,

We’re pleased to inform you that your manuscript has been judged scientifically suitable for publication and will be formally accepted for publication once it meets all outstanding technical requirements.

Kind regards,

Yung-Fu Chang

Academic Editor

PLOS ONE

Additional Editor Comments (optional):

Reviewers' comments:

Reviewer's Responses to Questions

**Comments to the Author**

1. If the authors have adequately addressed your comments raised in a previous round of review and you feel that this manuscript is now acceptable for publication, you may indicate that here to bypass the “Comments to the Author” section, enter your conflict of interest statement in the “Confidential to Editor” section, and submit your "Accept" recommendation.

Reviewer #1: All comments have been addressed

2. Is the manuscript technically sound, and do the data support the conclusions?

Reviewer #1: Yes

3. Has the statistical analysis been performed appropriately and rigorously? 

Reviewer #1: Yes

4. Have the authors made all data underlying the findings in their manuscript fully available?

Reviewer #1: Yes

5. Is the manuscript presented in an intelligible fashion and written in standard English?

Reviewer #1: Yes

6. Review Comments to the Author

Reviewer #1: All questions have been addressed adequately. I hope you continue doing research on leptospirosis, regarding differential diagnosis, immune pathogenesis and epidemiology.

7. PLOS authors have the option to publish the peer review history of their article (what does this mean?). If published, this will include your full peer review and any attached files.

Reviewer #1: **Yes: **Decio Diament

---

## [Editor Report · Acceptance letter]

25 Nov 2021

PONE-D-21-30348R1 

Clinical spectrum of endemic leptospirosis in relation to cytokine response 

Dear Dr. Agampodi:

I'm pleased to inform you that your manuscript has been deemed suitable for publication in PLOS ONE. Congratulations! Your manuscript is now with our production department. 

Kind regards, 

on behalf of

Dr. Yung-Fu Chang 

Academic Editor

PLOS ONE